# A Fully Wireless Wearable Motion Tracking System with 3D Human Model for Gait Analysis

**DOI:** 10.3390/s21124051

**Published:** 2021-06-12

**Authors:** Kevin Lee, Wei Tang

**Affiliations:** 1Engineering Physics and Electrical Engineering, New Mexico State University Main Campus, 1125 Frenger Mall, Las Cruces, NM 88003, USA; kevinlee@nmsu.edu; 2Klipsch School of Electrical and Computer Engineering, New Mexico State University Main Campus, 1125 Frenger Mall, Las Cruces, NM 88003, USA

**Keywords:** interlimb coordination, IMU, gait analysis, motion tracking, wearable devices

## Abstract

This paper presents a wearable motion tracking system with recording and playback features. This system has been designed for gait analysis and interlimb coordination studies. It can be implemented to help reduce fall risk and to retrain gait in a rehabilitation setting. Our system consists of ten custom wearable straps, a receiver, and a central computer. Comparing with similar existing solutions, the proposed system is affordable and convenient, which can be used in both indoor and outdoor settings. In the experiment, the system calculates five gait parameters and has the potential to identify deviant gait patterns. The system can track upper body parameters such as arm swing, which has potential in the study of pathological gaits and the coordination of the limbs.

## 1. Introduction

Human motion analysis is one of the most important tools and biomarkers to study pathological gaits, such as Traumatic Brain Injury (TBI) [1,2,3] and overall health in elderly population [4,5]. Currently, many medical providers perform gait analysis using only their own observations and rudimentary tools such as a stopwatch. An autonomous human motion monitoring analysis system would be beneficial since it can perform playback from different angles, which allows medical providers to discern smaller perturbations that the human eye cannot detect at a first glance. Moreover, an autonomous gait analysis system can provide deeper analyses to help medical providers make decisions. Currently, the main challenge of deploying such autonomous motion-sensing systems comes from complexity, convenience, and cost. The users are expected to have low-complexity devices without technology intensive tools such as pressure mats, webcams, magnetic tracking, or will mote controls. This is because the users who are usually mentally stressed would appreciate only the simple to operate devices [6]. For example, a camera-based motion capturing system requires complicated procedures to set up. It is also inconvenient since it requires a fixed arena, which does not allow outdoor or at-home testing. Moreover, the cost of such a system is very high, which could be more than $200k.

Wearable sensors [7,8] can provide low-cost automatic monitoring and processing of both motion [9,10] and physiological signals [11], and is capable of identifying abnormal signals. Compared with other motion sensing devices such as treadmills, wearable motion sensors can record detailed limb movement of the individual patient and provide comprehensive kinematic data. Moreover, the wearable sensor system has the potential of performing analysis of the abnormality of its recorded data by comparing both a well-matched reference from the able-bodied group database and the individual patient’s historical data, in order to achieve the goal of patient-specific evaluation. Furthermore, the wearable motion sensors are easy to use, which can be prescribed to the patient for at-home applications without clinical equipment such as a camera-based recording system. Nevertheless, the technical challenge of realizing such a system comes from both the hardware and the software. For example, continuous data recording and analyzing require power-efficient wireless sensor hardware [12,13,14]. In addition, the real-time high-throughput data interface expects algorithms that can handle a large amount of data with low computing overhead. Third, the system should provide a friendly interface to the medical providers such as 3D views and playback of the patient movement from different angles, especially for pathological gait analysis.

There has been an interest to study interlimb coordination for pathological gait analysis. For example, asymmetrical gait patterns of lower extremities [15,16] and upper extremities [17] have been studied. Interlimb coordination helps minimize energy consumption, decrease vertical ground reaction forces (GRFs), optimize stability, and optimize neural performance. Interlimb coordination focuses on both the angle and phase symmetries between the left and right sides of the body, as well as between the upper and lower limbs. Currently, such studies usually apply the high-speed camera system, which is not convenient for clinical studies. Also, the extremities do not contain enough information to study the interlimb coordination since the data from the elbows and knees are missing. Therefore, wearable motion sensors that could provide more data from every joint of each limb are expected to advance the study of interlimb coordination. The system should be able to report the angle and phase data of limb coordination, perform analysis in real-time, and be able to provide a playback of the subject from different angles. So the medical providers can make decisions based on both the visual observation and automatic calculation provided by the system since it is critical to keep humans in the loop for making medical decisions.

The main contribution of the paper including (1) Full body wireless wearable motion-sensing system, (2) A real-time human model that tracks the subject’s movement with playback function in the graphical user interface, and (3) gait analysis and interlimb coordination study using the proposed system. This paper is organized as follows: Section 3 details the hardware and software components of the system. Section 4 describes the gait analysis using the proposed system. Section 5 presents the experimental results. Section 6 concludes the paper.

## 2. Related Work

Currently, most IMU-based gait analysis systems use less than six sensors which were placed either on the feed or shank [18,19,20]. The most common methods are to use Euler angles (yaw, pitch, roll), angular velocity, linear acceleration, and quaternions. A comprehensive review of over 40,000 papers related to gait analysis can be found in [21]. The studies of inter-limb coordination often focus on asymmetrical gait patterns of the lower extremities [15] or upper extremities [17]. For instance, [16] studied the effects of small perturbations and phase response curves for both legs using high-speed cameras without IMUs. The market-available wearable motion-sensing systems are not ideal for studying the interlimb coordination of pathological gait for several reasons. First of all, most of the currently reported wearable sensors are focusing on record the motion of individual limb segment, which are not ideal for the study of interlimb coordination. For example, the APDM Opal sensor [22] uses six sensors, since there was only one sensor on each limb, such a system would not be able to precisely calculate the angles of shoulders, elbows, knees, and ankles. Therefore it could not be applied for the study of interlimb coordination. Other typical systems such as [23] only cover the upper body and [24] only focuses on lower limbs. Another problem is that the current systems [23,25] are emphasizing parameter calculation and waveform recording of individual sensors, which do not provide a real-time reconstruction of human motion with playback function for medical providers. The third problem is that the high-performance fully wireless real-time recording systems are very expensive, such as the XSense system, which makes it difficult for basic research projects. Therefore, a wireless wearable motion-sensing system is expected to study interlimb coordination with low-cost, full-body monitoring with all limbs, and the function of 3D human model reconstruction and playback.

To address the aforementioned problems, we developed a fully wireless wearable motion-sensing system using inertial measurement units with Zigbee wireless transceiver for data communication. The system contains ten sensors, two for each upper limb and three for each lower limb, to track the movement of individual segments of each limb. A graphical user interface has been designed so that a 3-dimensional 360-degree human model can track the subject’s movement in real-time. The medical providers can use playback functions to view the limb movement from any angle. Quaternion calculations are applied in limb orientation calculation to avoid gimbal lock problems from the Euler angle method. The system can perform gait analysis and be applied in the study of interlimb coordination. The system provides a convenient tool for both the patient and the medical providers to study gait, kinematic, and kinetic variables. The human model also helps in providing privacy compared to the camera-based systems.

## 3. System Design

Our proposed full-body wireless motion-sensing system consists of the hardware sensing and wireless communication system, the software data acquisition, and display system. The hardware system consists of ten wireless sensors with straps, one receiver, and a laptop to process the data and control the system. The software system includes the data collection and display function, the control function, and the signal processing functions for gait analysis. The overall system is shown in Figure 1, where Figure 1a illustrates the ten sensors on the subject while Figure 1b shows the visualization of the reconstructed 3D location of the limbs from the software system. The sensor and receiver circuit boards are shown in Figure 1c,d, respectively. The ten sensors are placed on the front arms, upper arms, thighs, shanks, and feet. Each sensor has a unique sensor ID, records the rotation of the corresponding limb, and sends the data wirelessly to the central receiver. A computer program collects data from each limb and reconstructs the full-body movement. Gait parameters are then extracted and processed to identify the abnormality of the limb movements. The following subsections present the system design and implementation details.

### 3.1. Hardware Design

Each of the ten wireless sensors in the system consists of a printed circuit board that hosts a battery, a microprocessor board, a sensor board, and a wireless module. The wireless sensor is placed in a small plastic bag attached to an athlete strap. A push-button switch controls the power supply of the system. The schematic of the sensor circuit is shown in Figure 2. Each sensor uses a 3.7 V 500 mAh Lithium Ion Polymer Battery. The batteries can last more than six hours of continual use. The battery pins are connected to the switch, which is connected to the power supply pin of the microcontroller. The microcontroller supplies power of 3.3 V to the sensor and the wireless unit. The sensor provides data to the microcontroller using a 2-wire Serial Bus with one Clock line and one Data line. Then the microcontroller organizes the input data from the sensor and sends the wireless packet data to the wireless unit using 2-pin (input and output) data interface. The wireless unit then transmits the data packet wirelessly to the receiver, where data from each sensor are combined for reconstructing the 3D model and performing gait analysis.

The sensor used in this work is the BNO055 inertial measurement unit, which contains an accelerometer, a magnetometer, and a gyroscope. The sensor calculates its orientation in real-time and sends the data to its host microcontroller. The sensor uses an I2C bus with one data pin (SDA) and one clock pin (SCL) to collect the IMU data to the host microcontroller. The BNO055 sensor can provide data with 9 degree-of-freedom (DOF), including the orientation, angular velocity, acceleration, gravity, and magnetic vectors, and temperature. In this design, we only used the quaternion output data of the absolute orientation. Each of the ten sensors in the system is set to local instead of global orientation mode through the host microcontroller. In this mode, the sensor does not align itself with magnetic north. Instead, each sensor makes measurements relative to the orientation it starts. In the operation, each sensor allows a delay of 20 ms to avoid overloading the host microcontroller. The maximum sampling rate of the sensor is 100 Hz. However, in our application, the sampling rate is set at 59 Hz to ensure that 10 sensors can work together. 59 Hz is also the maximum reliable sampling frequency that the system can achieve. The BNO055 sensor includes internal algorithms to constantly calibrate the gyroscope, accelerometer, and magnetometer inside the device. The sensor starts supplying sensor data as soon as it is powered on. The sensors are factory trimmed to reasonably tight offsets, meaning we can get valid data even before the calibration process is complete. The calibration automatically starts once the sensor finds magnetic north. Once the device is calibrated, the calibration data are kept until the BNO is powered off.

Each sensor is equipped with a microcontroller to collect the data from the IMU and send the data into the wireless unit. Specifically, each time the sensor senses the data, all the data from the IMU are sent to the microcontroller through the I2C bus. The microcontroller only selects the Quaternion data of the absolute orientation and writes the data to the wireless unit using the transmitter TX pin and the receiver RX pin. Each quaternion data packet begins with the sensor ID number from 0 to 9, with the following values of *w*, *x*, *y*, and *z* data. By doing so the received data can be categorized by the sensor ID numbers for the 3D reconstruction of the limb angles. The microcontroller in our design is the Adafruit Pro Trinket board which utilizes the Atmel ATtiny85 chip. The microcontroller is programmed using the Arduino libraries especially the Unified Sensor Library and the BNO055 library.

The wireless units in the system are the XBee trace antennas that compatible with IEEE 802.15.4 standard for low-power and short-distance data communication. The measured transmission distance of the wireless units is 30 m. A total of 11 XBee antennas are used in the system for 10 sensors and one receiver. Each XBee antenna is coupled with an XBee Explorer Regulated breakout board, which performs power regulation and signal conditioning. The 10 sensor antennas are configured as endpoints while the receiver antenna is configured as the coordinator. This forms a Star Network as shown in Figure 3. Since the 10 sensors are sending data to the receiver simultaneously, the Carrier Sense Multiple Access/Collision Avoidance (CSMA/CA) mechanism is applied in the network. Under CSMA/CA, the XBee transmitter on the sensor has to monitor the channel to check for any activity on the channel. The transmitter on the sensor starts to transmit the signal only when the channel is “idle”, or else the sensor has to defer the transmission. Each sample has its timestamp. So even if some samples are lost the system is still able to track the limb movement.

The hardware circuits of the wireless units for the sensor and for the receiver are identical. Each XBee trace antenna was set up using XCTU software by Digi to set the PAN ID, Source Address, and the baud rate. The PAN ID is the same for each antenna while the source address is different for each antenna. The baud rate is set at 115,200. The Coordinator Enable setting in XCTU determines the device to be either an “End Device” for the sensor or a “Coordinator” for the receiver. The receiver is connected to a computer for data collection, visualization, and processing. The program on the computer is coded using Processing with JavaScript. The program reads the serial communication from the receiver and categorizes the quaternion data based on the unique Sensor ID. A graphical user interface is then activated for visualization, which reconstructs the limb motions in real-time. The total hardware cost of this system including ten sensors and the receiver is $603.33. However, the price can be significantly reduced if a different orientation sensor is used. It may have been more cost-effective to use another sensor because the BNO055 incurred the highest cost for this project at a rate of $34.95 per unit. The cost of our system is much lower than the market available solutions which are usually above $3000 [26].The data collection and visualization procedure is presented in the next subsection.

### 3.2. Software Design

The software component of the system including data acquisition, 3D reconstruction, and visualization. The data acquisition software contains the Arduino code implemented on the microcontroller on the wearable sensor, which is programmed in C language, and the Processing code implemented on the receiver, which is programmed in Javascript. The receiver sends the data to the computer using the serial communication port. A graphical user interface (GUI) is developed to monitor the data communication and plot the 3D reconstruction of the limbs for visualization. The GUI also provides a control interface to the user and the record/playback functions to create a friendly environment for the medical provider to access and evaluate the gait data. The 3D reconstruction and visualization are also programmed with Processing using Javascript. Special efforts have been made to reconstruct the 3D orientations of the 10 segments in the four limbs using quaternion data, for example, combining segments to form a limb while avoiding gimbal lock. This subsection focuses on the algorithm of data acquisition, visualization, and limb orientation calculation.

The data acquisition and visualization system use the Serial and Open Graphics Library (OpenGL) with JavaScript APIs. Human limbs are modeled using a simple ball and sticks model, in which the balls are modeling the joint and the sticks are modeling the limb segments. Since the IMU tracks the absolute orientations, the system can monitor the limbs beyond 360 degrees. This means that the subject wearing the sensor can continuously spin the arms in a circle and the system is still able to track the orientation of the arm. Since only orientations are monitored, the human model is simplified by setting segments of the arms and legs with fixed lengths. Each limb is set at the origin point (0, 0, 0) and then is translated to their respective positions for the real-time visualization. For visualization, a camera function was implemented so that the digital human model can be seen from all different angles. The user can rotate the camera clockwise or counter-clockwise from the GUI. During initialization, each limb starts pointing down with a starting angle of 0 degrees.

During data acquisition, the receiver collects the quaternion data (*w*, *x*, *y*, *z*) from the 10 sensors to generate a *n*x40 matrix. Here *n* is the number of data points collected and the 40 columns are the quaternion value from the 10 sensors. The system sampling frequency is 59 Hz, so a 10-s recording generates 590 rows of the data. The BNO055 sensor uses a signed 2-byte data format to store quaternion data. In our design, each quaternion value is stored up to four decimal places. The data can be used to generate a 3D real-time reconstruction of the human motion while being stored in a text file. After recording, the GUI provides a function to load the stored data so the recorded motion can be replayed. The medical provider can also change the angle of view during recording and replay. This feature provides convenience for medical providers to exam the gait trial at different angles, which brings advantages over the methods of video recordings.

The quaternion data from each sensor are combined to calculate the limb orientation for the 3D human model. In the 3D human model, each arm is made up of two segments (upper arm and front arm) and each leg is made of three segments (thigh, shank, and foot). Each segment’s orientation is calculated individually and then placed together in the visualization program. Figure 4b graphically shows an example of the process to connect two segments to form an arm. In this case, segment 1 is used to represent the upper arm and segment 2 represents the front arm. Both segments initially start at the origin in the 3D space. Once the orientation of each segment is obtained from the sensor, segment 2 is translated to the end of segment 1 to form the whole arm. In our design, each limb is reconstructed once all the segment data are received. If one segment is missing, the whole limb can not be reconstructed.

## 4. Gait Analysis

The system calculates five gait parameters: stride length, gait speed, cadence, double support, and swing phase [27]. The stride length is the distance between the successive heel contact points of the same foot. Normally, stride length is twice the step length. A step length is a distance between the heel contact point of one foot and that of the other foot. The gait speed is the time it takes to walk a specified distance. The cadence is the rate at which a person walks, expressed in steps per minute. The double support is the phase in which the bodyweight is supported by both legs. Double support is measured from initial contact to the end of loading response. The swing phase is the part of the gait cycle that lasts from toe-off to the next initial contact.

If we take the time between the two events we can calculate double support using Equation (Equation 1).
(1)$$\mathrm{Double}\;\mathrm{support}=\frac{\sum_{i=1}^{n-1}T\left(i\right)-H\left(i\right)}{T_{total}}$$
where *n* is the number of peaks, *T* is the row vector containing the times of opposite toe off, *H* is the row vector containing the times when initial heel contact occurred, and $T_{total}$ is the total duration of the trial.

In order to calculate stride length, the user needs to input their leg length (*l*). The stride length is calculated using Equation (Equation 2)
(2)Stridelength=lsinH(i+1)+lsinH(i)

Cadence is calculated by counting the number of peaks and dividing it by two. For gait speed, the user needs to input the distance that they traveled (*d*) using Equation (Equation 3).
(3)$$\mathrm{Gait}\;\mathrm{Speed}=d/T_{total}$$

Swing phase is calculated using Equation (Equation 4)
(4)SwingPhase=H(i+1)−T(i)

The proposed system calculates these five gait parameters from knee extension and flexion data. In our simplified gait model, we assume that initial heel contact occurs when the above-knee angle is maximized, and we assume that toe-off occurs when the angle is minimized.

## 5. Experimental Results

Ethical approval for this work has been granted by the Office of Research Compliance of New Mexico State University with project ID 20292. The recorded knee angle data during one experimental trial of gait analysis is shown in Figure 5. The benefit of using these methods is that we do not need exact knee flexion and extension angles to perform gait analysis. For example, it is possible that the system was not properly calibrated before use, so the angle data may be off by several degrees. However, since the actual angle value is not used in the gait analysis, this does not affect the results since only the timing of the maximum and minimum knee angles are applied instead of the exact angle values. We have two subjects tested the system to collect the data with two runs. Figure 5 measures the angles and peaks of the two shank sensors that are used to measure the gait parameters. Our algorithms first find the peaks of both the left leg and the right leg. Then the algorithm calculates the gait parameters using the timing information and the measured leg length of the subject. The average calculated gait parameter from the system is summarized in Table 1 comparing with the average known data. The system errors are analyzed in the following discussions.

The average error of IMU for measuring the angles is between 6% to 10% [23]. Besides, the location of the sensors is another source of error. In the application, straps that hold the sensors can be placed very loosely causing additional movements which cause inaccuracies in the gait analysis. In addition to these systematic errors, uncertainties can arise from the footwear of the patient. For example, someone’s gait may be entirely different while wearing high heels versus sneakers. A table comparing the performance between the proposed system in this paper and other recent similar systems is summarized in Table 2.

In the experiment of interlimb coordination, the subject performs simple walking while wearing the sensors. The sensing system records the angles with respect to time from arms, shoulders, hips, knees, shanks, and feet. Then the software calculates the cross-correlation between the angle waveforms. The box plot of the cross-correlation coefficient is shown in Figure 6. The results show that if the data are from the same sample (same subject during the same trial), the correlation coefficient is higher, which means that the limb angles data are correlated to each other. Otherwise, the correlation coefficient is much lower, which means the angle data are not correlated. This can be used to evaluate if the limbs are coordinating during a trial.

## 6. Conclusions

In this communication, we reported a motion tracking system that monitors both the upper and lower limbs. The system is fully wireless and uses quaternion data. The graphical user interface can reconstruct the 3D human model in real-time at a sampling frequency of 59 Hz. Playback and recording features were incorporated into the interface. The experimental results report five gait parameters calculated using the proposed system. The system provides a low-cost and convenient tool for pathological gait monitoring and analysis.

## Figures and Tables

**Figure 1 sensors-21-04051-f001:**
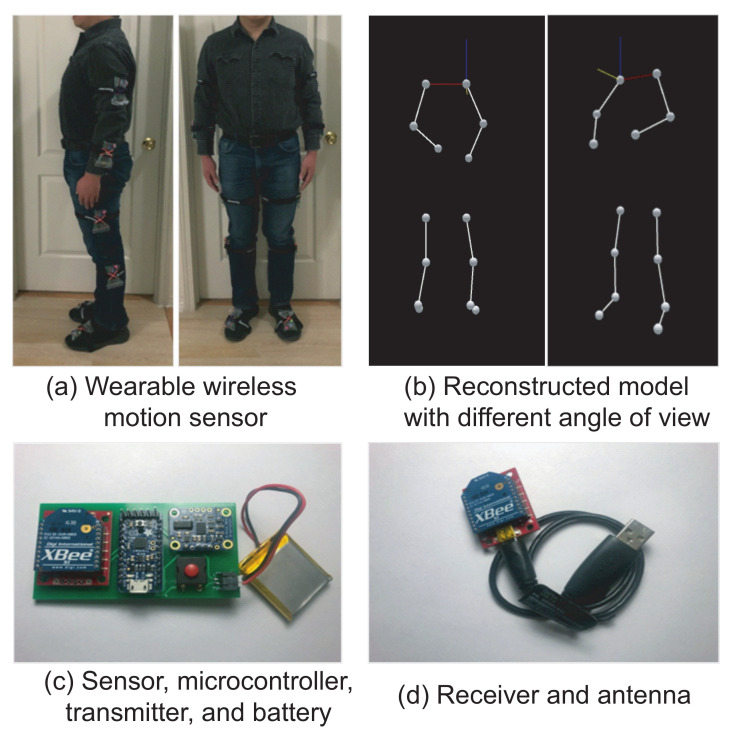
The proposed fully wireless wearable motion sensors and the real-time reconstructed 3D human model.

**Figure 2 sensors-21-04051-f002:**
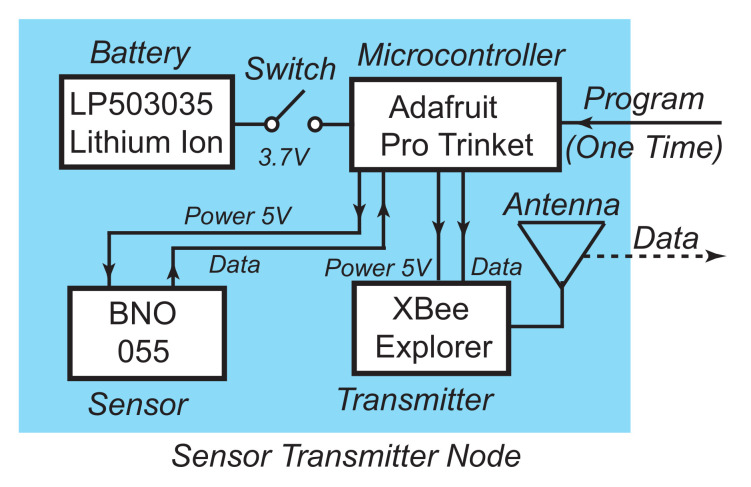
Simplified block diagram of the wireless motion sensor unit.

**Figure 3 sensors-21-04051-f003:**
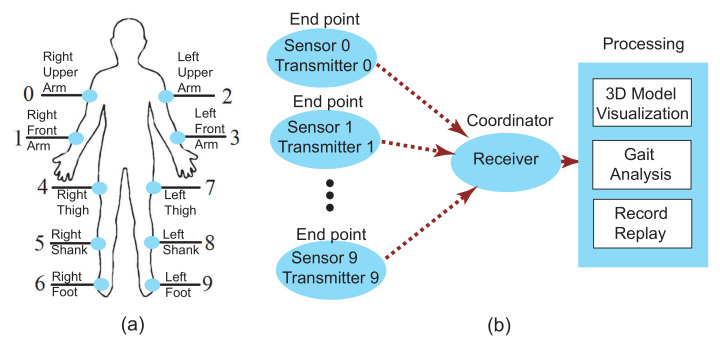
The sensor deployment (**a**) and the star network configuration of the fully wireless motion sensing system (**b**).

**Figure 4 sensors-21-04051-f004:**
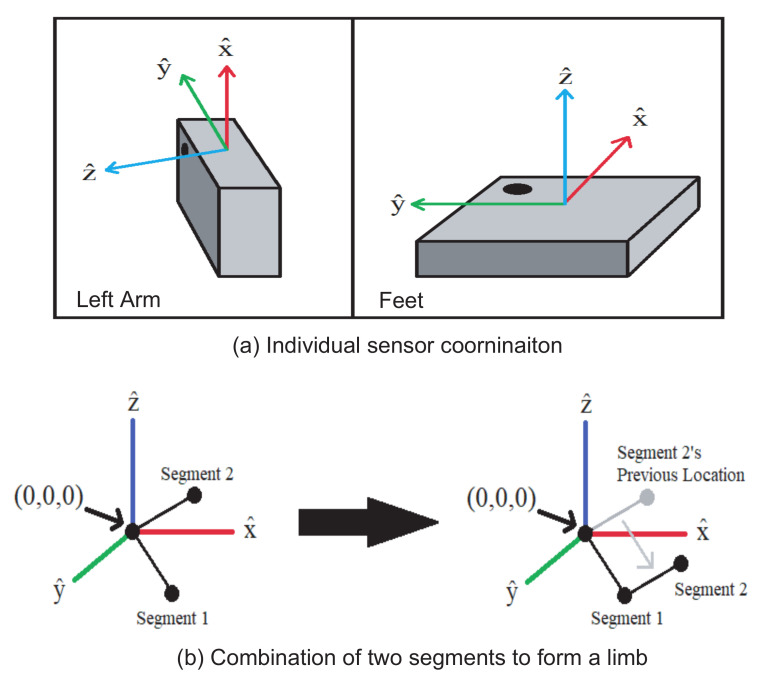
Sensor coordinate and segment combination for limb construction in the 3D human model.

**Figure 5 sensors-21-04051-f005:**
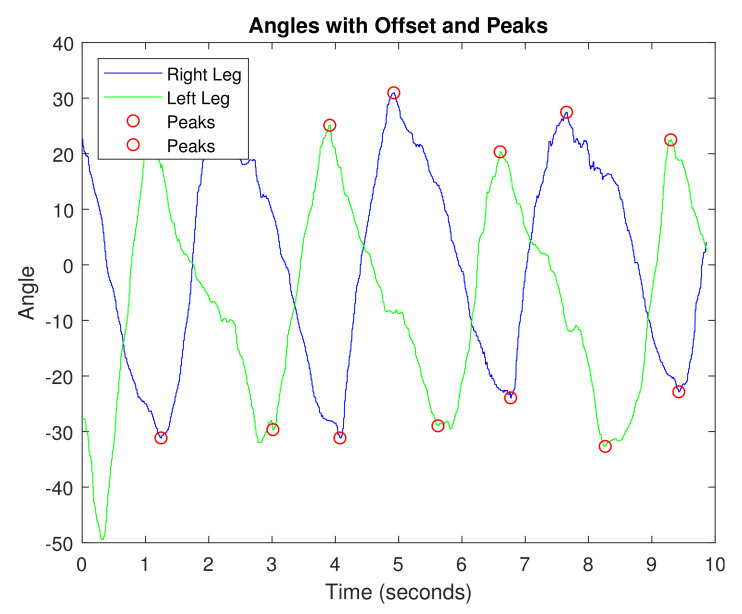
The recorded knee angle data for gait analysis.

**Figure 6 sensors-21-04051-f006:**
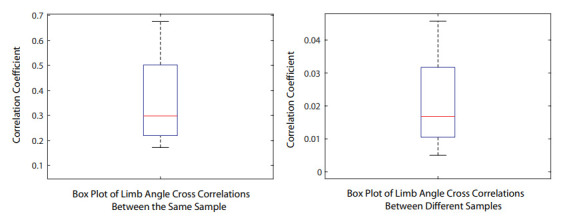
Interlimb cross-correlation results from the same trial (**left**) and different trials (**right**). The correlation coefficient shows that the limbs coordinates in the same trial. The cross-correlation results can be used to classify if the limbs are coordinating during walking, which would indicate abnormal gaits.

**Table 1 sensors-21-04051-t001:** The calculated gait parameter comparing with the known value.

Parameter	Sys. Calculated	Known Value
Stride Length	101 cm	120 cm
Gait speed	44 m/min	50 m/min
Cadence	73.7 steps/min	80 steps/min
Double support	0.84	0.80
Swing Phase (L)	0.34	0.40
Swing Phase (R)	0.38	0.40

**Table 2 sensors-21-04051-t002:** Performance comparison between the proposed system and recent reported similar systems.

	This Work	[28]	[23]	[22]
		2015	2020	2017
Target Application	Whole body motion capture	Whole body motion capture	Motion capture of upper extremity	Whole body motion capture
Number of Sensors	10 IMU Full Body	N/A	5 Upper Body Only	6 IMU Full Body
Inter Limb Coordination	Yes	Yes	No	No
Sampling Frequency or Processing Time	59 Hz Real-Time	N/A	92.5 μs Computing time	N/A
3D Human Model	Yes	Yes	No	Yes

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
