# Peer review of "A Fully Wireless Wearable Motion Tracking System with 3D Human Model for Gait Analysis"

_sensors, 2021, doi:10.3390/s21124051_

Round 1

Reviewer 1 Report

The authors address the research problem of tracking and analyzing human gait. The primary motivations presented by the authors are related to the autonomous motion-sensing systems' complexity, convenience, and costs. The considered approach uses a set o ten sensors attached to different parts of the human body.  Still, according to the authors, the proposed solution is to reduce fall risk and retrain gait in a rehabilitation scenario.

Generally, the text is well-written; however, the authors must revise some text parts (e.g., places providing a list of items, with extensive use of "and" word to separate them without any comma). In addition, there are other places where a word appears to be missing.

The "Introduction" section was mixed with related work analysis. My suggestion is to separate them into appropriate sections: Introduction and then "Related Work". Besides that, the Introduction could be well elaborated, with a better description of the research's main challenge (using some paper references to justify why "complexity, convenience, and costs are the main hurdles of this topic). 

The results presented in the end seem to be incipient. How many runs were executed to infer the presented results? How many end-users wore the solution? 

Additionally, the authors say that: "Comparing with similar existing solutions, the proposed system is affordable and convenient, which can be used in both indoor and outdoor settings". I would expect to see some cost comparison in the paper, mainly to justify this affordability statement (once the proposed system uses 10 sensors and additional hardware to process the data).

Next, the authors say that one of this paper's contributions is: "... (3) gait analysis and interlimb coordination study using the proposed system". Section 3 barely provides the gait analysis. However, I miss where "interlimb coordination study" is addressed. I recommend that the authors stress this data while analyzing other variables other than knee angles.

Still, there is no mention of any calibration procedure. When analyzing a solution of this kind, it is expected to see some calibration steps before any data gathering. Indeed, the calibration could even help to reduce the error rate presented in the results section.

The authors also state that: "Gait parameters are then extracted and processed to identify the abnormality of the limb movements". My question here is: How is it done? Which are the steps taken into account to identify these abnormalities? As a suggestion, I would like to see a flow graph showing the steps in which the work proceeds to verify if a movement is abnormal or not.

In the way the research is, it presents some contribution to the area, especially when considering the context of this journal issue. However, the results presented in the end show a reasonable level of errors when comparing the solutions' data with real values. Therefore, my suggestion here is to investigate the errors better and mitigate them to create a solution close to the real world. Additionally, some more experimental data should be collected and analyzed to justify the results better, supporting the paper affirmations. 

Reviewer 2 Report

[1] There are some places where the description in the figure and the text do not match.

(i) The captions of Fig.1(a)(b)(c)(d) do not match the description in line 109 to 110.

(ii) Line 125 states that the microcontroller supplies power of 3V to the sensor and the wireless unit, but it is 5V in Fig.2. In actual, the supply voltage of BNO055 and XBee must be 3V.

(iii) Lines 226-227 state that Fig.4 shows an example for an arm, but Fig.4(a) is the illustration of sensors of left arm and feet.

[2] In Fig.2, the directions of signals between the microcontroller and XBee are both downward.

[3] Please explain the rationale for setting the sampling frequency to 59Hz. Does the number of 59 mean anything?

[4] Lines 156-157 state that the serial communication between the microcontroller and the sensor is set at a baud rate of 115200, but it is the communication between the microcontroller and XBee that is 115200baud serial communication.

[5] According to the description in Lines 158--170, the order of data transmission from 10 sensor units to host computer is not controlled by the host computer. How do you ensure the synchronization of data sampling in each sensor unit. At 115200baud, it probably takes about 1ms for the data transmission of one unit. The time margin is only about 7ms for the period of 59Hz. Explain that it is always possible that all data can received in a period of 59Hz.

[6] In Section 4, please describe the explanation of the experimental data in more detail.

Reviewer 3 Report

The manuscript is a communication on an “IMU-based motion tracking system for gait analysis”. Major revisions are needed.

Although the topic is very interesting, I don’t understand why the authors are presenting a short communication and not an original research article.

Moreover, the market has already several IMU-based motion capture systems. What is the added value of your system?

Please add more details on the costs.

What could be the clinical impact of this system?

Round 2

Reviewer 1 Report

The authors addressed most of the revised points. Thus, these corrections improved manuscript quality and clearness. 

Additionally, I would like to suggest final proofreading to fix minor points. For instance:
  - Line 186: The serial baud rate speed is listed as 115220 (instead of 115200 kbps);
   - Line 197: Please, provide the reference for the concurrent solution that costs $2000. 

Anyway, these are minor changes should not prevent the manuscript public

Author Response

Thanks for your notes. The baud rate is set at 115200. The price reference has been added to the reference.

Location: Line 187, Ref [26].

Reviewer 2 Report

In Fig. 6, you can see the difference better by unifying the scales on the vertical axis of the two graphs. 

Author Response

We tried to modify Fig. 6 using the same scale of the two plots, however, since the right plot is much lower it looks not clear. So we decided to keep the current format of Fig. 6.

Reviewer 3 Report

The author revised the manuscript and answered all my comments and suggestions.

The paper is not acceptable for being published.

Author Response

Thanks for your review. Based on the review comments, I guess the word "not" should be "now".